# Circulating Myeloperoxidase (MPO)-DNA complexes as marker for Neutrophil Extracellular Traps (NETs) levels and the association with cardiovascular risk factors in the general population

**Samantha J. Donkel**[ID][1], **Frank J. Wolters**[ID][2,3], **M. Arfan Ikram**[2], **Moniek P. M. de Maat**[1] *

**1** Department of Hematology, Erasmus University Medical Center, Rotterdam, The Netherlands,
**2** Department of Epidemiology, Erasmus University Medical Center, Rotterdam, The Netherlands,
**3** Department of Radiology & Nuclear Medicine, Erasmus University Medical Center, Rotterdam, The Netherlands

* s.donkel@erasmusmc.nl

**Data Availability Statement:** All relevant data are within the paper and its Supporting Information files.

## Abstract

### Introduction

Neutrophil extracellular traps (NETs) are DNA scaffolds enriched with antimicrobial proteins. NETs have been implicated in the development of various diseases, such as cardiovascular disease. Here, we investigate the association of demographic and cardiovascular (CVD) risk factors with NETs in the general population.

### Material and methods

Citrated plasma was collected from 6449 participants, aged ≥55 years, as part of the prospective population-based Rotterdam Study. NETs were quantified by measuring MPO-DNA complex using an ELISA. We used linear regression to determine the associations between MPO-DNA complex and age, sex, cardio-metabolic risk factors, and plasma markers of inflammation and coagulation.

### Results

MPO-DNA complex levels were weakly associated with age (log difference per 10 year increase: -0.04 mAU/mL, 95% confidence interval [CI] -0.06;-0.02), a history of coronary heart disease (yes versus no: -0.10 mAU/mL, 95% CI -0.17;-0.03), the use of lipid-lowering drugs (yes versus no: -0.06 mAU/mL, 95% CI -0.12;-0.01), and HDL-cholesterol (per mmol/l increase: -0.07 mAU/mL/, 95% CI -0.12;-0.03).

### Conclusions

Older age, a history of coronary heart disease, the use of lipid-lowering drugs and higher HDL-cholesterol are weakly correlated with lower plasma levels of NETs. These findings

**Funding:** The Rotterdam Study is supported by the Erasmus MC University Medical Center and Erasmus University Rotterdam; The Netherlands Organisation for Scientific Research (NWO); The Netherlands Organisation for Health Research and Development (ZonMw); the Research Institute for Diseases in the Elderly (RIDE); The Netherlands Genomics Initiative (NGI); the Ministry of Education, Culture and Science; the Ministry of Health, Welfare and Sports; the European Commission (DG XII); and the Municipality of Rotterdam. We acknowledge the support of the Netherlands Cardiovascular Research Initiative which is supported by the Dutch Heart Foundation (CVON2015-01: CONTRAST), the support of the Brain Foundation Netherlands (HA2015.01.06), and the support of Health~Holland, Top Sector Life Sciences & Health (LSHM17016), Medtronic and Cerenovus. The collaboration project is additionally financed by the Ministry of Economic Affairs by means of the PPP Allowance made available by the Top Sector Life Sciences & Health to stimulate public-private partnerships. The measurement of MPO-DNA complex levels in participants of the Rotterdam study was supported by a research grant (Prof. Heimburger Award 2018, CSL Behring). The funders had no role in study design, data collection and analysis, decision to publish, or preparation of the manuscript.

**Competing interests:** This study was supported by the commercial funders: Health-Holland, Medtronic and Cerenovus. This does not alter our adherence to PLOS ONE policies on sharing data and materials.

show that the effect of CVD risk factors on NETs levels in a general population is only small and may not be of clinical relevance. This supports that NETs may play a more important role in an acute phase of disease than in a steady state situation.

## Introduction

Neutrophils contribute to host defense through different mechanisms, including the formation of neutrophil extracellular traps (NETs) [1]. The process of NET formation is a relatively recently identified form of cell death by neutrophils which was first described by Brinkmann et al. in 2004 [2]. NETs are formed when neutrophils secrete decondensed intracellular DNA together with antimicrobial proteins such as myeloperoxidase (MPO) and neutrophil elastase. This forms a web-like structure where pathogens can be trapped and killed [2]. In addition to their function in immunity, NETs have been implicated in the pathophysiology of thrombosis [3], atherosclerosis [4], and sepsis [5, 6]. For example, NETs are found in the thrombus of patients with myocardial infarction and stroke, suggesting a role of NETs in thrombus formation [7, 8]. A better understanding of the influence of NETs in the development of diseases requires knowledge about the demographic and cardiovascular determinants (e.g. age, sex, medical history, blood markers of inflammation and coagulation) of NET formation in a general population.

Some of the mechanisms by which NET formation is determined have already been investigated in *in vitro* and *in vivo* studies [9–12]. For example, studies in mouse models and *in vitro* studies in human neutrophils have shown that the amount of NET formation decreases with increasing age. Furthermore, sex differences in NET levels have been described in patients with multiple sclerosis, where higher levels of NETs were found in males than in females [12]. Also, in neutrophils isolated from nondiabetic human subjects, high levels of glucose have been shown to induce NET formation [11]. To date, most studies on NETs levels in plasma, are case-control studies in patients with active underlying diseases where NETs levels are generally high. In order to gain more insight into biological processes involved in NET formation in a stable situation, without massive inflammation, it is important to look at determinants of NETs in the general population, where individuals are mostly free of acute diseases. Currently, population studies on NETs levels are lacking.

We therefore aimed to answer the following question: which demographic and cardiovascular risk factors are associated with NET formation in the general population?

## Materials and methods

### Study design and study population

This study is embedded in the Rotterdam Study, a prospective population-based cohort study among individuals of 55 years and older who are living in Ommoord, a suburb of Rotterdam, The Netherlands [13]. The original cohort started in 1990 (RS-I) and of the 10,215 eligible individuals, 7,983 agreed to participate. In 1999 the study was extended with 3,011 individuals (out of 4,472 invitees) who moved into the study district or reached the age of 55 years (RS-II). Participants visit the study center every 4 years for interview and extensive clinical assessment, including venipuncture and assessment of cardiometabolic risk factors. The Rotterdam Study has been approved by the Medical Ethics Committee of the Erasmus MC (registration number MEC 02.1015) and by the Ministry of Health, Welfare and Sport of the Netherlands

(Population Studies Act: WBO, license number 1071272-159521-PG). The Rotterdam Study has been entered into the Netherlands National Trial Register (NTR; www.trialregister.nl) and into the WHO International Clinical Trials Registry Platform (ICTRP; www.who.int/ictrp/network/primary/en/) under shared catalog number NTR6831. All participants provided written informed consent to participate in the study and to have their information obtained information from their treating physicians.

For this study, we used the data of the participants in the third examination of the original cohort (RS-I-3) between 1997 and 1999 (n = 4797) and the first examination of the extended cohort (RS-II-1) between 2000 and 2001 (n = 3011). We included all participants of whom blood samples were available (n = 6449).

## Population characteristics

Data of all participants was collected by structured interviews and physical examination. Blood samples were available of 6449 individuals. Blood pressure was measured as the mean of two readings using a random-zero sphygmomanometer in sitting position. We defined hypertension as a systolic blood pressure of 140 mmHg or higher, or a diastolic blood pressure of 90 mmHg or higher, or the use of blood pressure lowering medication. Antithrombotic medication was defined as the use of vitamin K antagonists, platelet aggregation inhibitors, and direct thrombin inhibitors. Lipid lowering agents were defined as the use of any lipid modifying agent. Diabetes mellitus was defined as fasting serum glucose level $\geq$ 7.0 mmol/L and/or the use of blood glucose lowering medication [14]. Total cholesterol and high-density lipoprotein cholesterol were measured using an automated enzymatic procedure in mmol/l. Body mass index was calculated as the weight (in kilograms) divided by the squared height (in meters). Smoking status was defined as current or no smoking at baseline. Coronary heart disease (CHD) was defined as fatal and non-fatal myocardial infarction and other coronary heart disease mortality. This includes myocardial infarction, myocardial revascularization, CHD mortality and overall CHD [15]. Stroke was defined as a syndrome of rapidly developing clinical signs of focal (or global) disturbance of cerebral function, with symptoms lasting 24 hours or longer or leading to death, with no apparent origin other than vascular [16]. Cardiovascular disease (CVD) was composed of CHD and stroke.

## MPO-DNA complex measurements

Citrated plasma samples were collected at the third visit of RS-I and the baseline examination of RS-II, and stored at -80˚C. We determined NET formation by measuring MPO-DNA complexes with a capture ELISA as reported earlier [4]. We adjusted the commercial human cell death ELISA kit (Cell death detection ELISAPLUS, Cat. No 11-774-425-002; Roche Diagnostics Nederland B.V., Almere, The Netherlands). Briefly, as the capturing antibody, we used anti-MPO monoclonal antibody (clone 4A4, ABD Serotec, # 0400–002). Patient plasma was added in combination with the peroxidase-labeled anti-DNA monoclonal antibody (component No.2 of the commercial cell death detection ELISA kit; Roche, #11-774-425-002). The absorbance at 405 nm wavelength was measured using Biotek Synergy HT plate reader with a reference filter of 490 nm. Values are expressed as milli arbitrary units per milliliter (mAU/mL). mAU were defined based on a preparation of NETs. Neutrophils were isolated as described previously [17], from a healthy volunteer and NET formation was induced by adding 250 ng/mL phorbol 12-myristate 13-acetate (PMA) (stock 100 µg/mL in DMSO). After an incubation period of 4 hours, we assigned a value of 1000 mAU/mL to this preparation. Every ELISA plate had its own reference curve (S2 Fig in S1 File) which was composed of the calibration material which was stored in aliquots at -80˚C before use. A total of 167 96-wells ELISA

plates were used to measure MPO-DNA complex levels of all participants and two different reference materials were used. A new reference line was calibrated to the old one by measuring the new material several times on the reference curve of the old material, after which we assigned the value to the new calibration material. In addition, a high and low control sample were added to every individual ELISA plate. Blood samples were measured in duplicate. Coefficient of variation (CV) of the high controls was 14.5% and the CV of the low controls was 12.3%.

## Measurement of coagulation, inflammatory and immunology markers

ADAMTS13 activity was measured using the fluorescence resonance energy transfer substrate VWF73 (FRETS-VWF73) as previously described [18]. VWF antigen (VWF:Ag) levels were determined with an in-house enzyme-linked immunosorbent assay, using polyclonal rabbit anti-human VWF antibodies (DakoCytomation, Glostrup, Denmark) for catching and tagging. Fibrinogen levels were derived from the clotting curve of the prothrombin time assay using Thromborel S as a reagent on an automated coagulation laboratory (ACL 300, Instrumentation Laboratory). In a subset of 1208 participants of RS-I-3, an extended panel of inflammatory and immunology markers was measured as previously described, including complement, immunoglobulins, and cytokines [19].

## Statistical analysis

Normally distributed data were presented as mean and standard deviation (SD), not normally distributed data were presented as median and 25th-75th percentiles. Categorical data were presented as number and percentage. MPO-DNA complex levels were not normally distributed and therefore log-transformed. Differences of MPO-DNA complex levels in different age categories and MPO-DNA complex levels at different time points during the day were analyzed using the Kruskall Wallis-test. Spearman correlation was used to calculate correlations between MPO-DNA complex and clinical characteristics as well as inflammatory markers. Linear regression analysis was performed to determine the association between MPO-DNA complex and demographic and clinical characteristics (age, sex, cardio-metabolic risk factors, a history of cardiovascular disease and blood markers of inflammation and coagulation) and circadian rhythm of MPO-DNA complex levels. The analyses were repeated after adjustment for age, sex and leukocyte count. Subgroup analysis was performed in different age categories, in men and women separately and in participants with and without the presence of comorbidities, including current smoking, diabetes mellitus, hypertension and a history of CVD. Data were analyzed using IBM SPSS Statistics for Windows, Version 25.0 (Armonk, NY: IBM Corp). All statistical tests were two-tailed and a p-value of <0.05 was considered statistically significant.

## Results

### Age, sex, time point of blood collection and the association with MPO-DNA complex levels

All participants of whom blood samples were available, were included in this study (n = 6449). The median age of the total population was 68.6 years (25th-75th percentile 62.8–75.3 years) and 3633 (56.3%) participants were female (Table 1). Median MPO-DNA complex levels were 53 mAU/mL (42–87). NETs were weakly correlated with age ($R_S$ = -0.07, p<0.01, S1 Table in S1 File). We found a weak inverse association of MPO-DNA complex with age (decrease of lnMPO-DNA complex per 10 year increase 0.04 mAU/mL, 95% confidence interval (CI)

**Table 1. Baseline characteristics of the total cohort.**

| | N = 6449 |
|---|---|
| Age, years | 68.6 (62.8–75.3) |
| Female sex, n (%) | 3633 (56.3) |
| Current smoking, n (%) | 1111 (17.2) |
| BMI, kg/m² | 26.9 ± 4.0 |
| Systolic blood pressure, mmHg | 143.3 ± 21.3 |
| Diastolic blood pressure, mmHg | 76.8 ± 11.3 |
| Hypertension, n (%) | 4376 (67.8) |
| Diabetes Mellitus, n (%) | 751 (11.6) |
| History of CVD, n (%) | 623 (9.7) |
| • Prevalent CHD | 413 (6.4) |
| • Prevalent stroke | 254 (3.9) |
| Antithrombotic medication, n (%) | 1332 (20.7) |
| Lipid-reducing agents, n (%) | 816 (12.6) |
| Total cholesterol, mmol/L | 5.8 ± 1.0 |
| HDL, mmol/L | 1.4 ± 0.4 |
| Glucose, mmol/L | 6.0 ± 1.6 |
| CRP, mg/L | 1.8 (0.7–3.8) |
| Leukocytes ($*10^{-9}$/L) | 6.8 ± 1.9 |
| Fibrinogen, g/L | 4.0 ± 0.9 |
| Von Willebrand Factor, IU/mL | 1.20 (0.93–1.60) |
| ADAMTS13, % | 91.4 ± 19.9 |
| MPO-DNA complex, mAU/mL | 53 (42–87) |

Normally distributed data are presented as mean ± standard deviation (SD), not normally distributed data are presented as median and 25[th]-75[th] percentiles. Categorical data are presented as number and percentage. BMI: body mass index, CVD: cardiovascular disease, CHD: coronary heart disease, HDL: high-density lipoprotein, CRP: C-reactive protein, ADAMTS13: a disintegrin and metalloproteinase with a thrombospondin type 1 motif, member 13.

-0.06;-0.02, p<0.01) (shown in Table 2 and Fig 1). We found no association between sex and MPO-DNA complex levels. Blood samples were collected between 8 AM and 4 PM on weekdays. There was no significant diurnal variation in MPO-DNA complex levels (S1 Fig in S1 File).

## CVD risk factors and MPO-DNA complex levels

A total of 623 (9.7%) participants had a history of CVD, of whom 413 (6.4%) had coronary heart disease (CHD) and 254 (3.9%) had a stroke (Table 1). Lipid-lowering drugs were used by 815 (12.6%) individuals and antithrombotic medication by 1332 (20.7%). Mean total cholesterol was 5.8 ± 1.0 mmol/L and mean high-density lipoprotein (HDL) was 1.4 ± 0.4 mmol/L. Median MPO-DNA complex levels in presence and absence of CVD risk factors are presented in S2 Table in S1 File. Hypertension, diabetes mellitus and a history of CVD were more prevalent in participants aged >75 years than in the other age categories. NETs plasma levels were negatively correlated with hypertension, history of CHD, lipid-lowering drugs and HDL (S1 Table in S1 File). We found a weak inverse association between MPO-DNA complex levels and a history of CHD (β -0.10 mAU/mL, 95% CI -0.17;-0.03), the use of lipid-lowering drugs (β -0.06 mAU/mL, 95% CI -0.12;-0.01), and HDL (β -0.07 mAU/mL/ mmol/L, 95% CI -0.12;-0.03) (Table 2). Adjustments for age and sex did not change the results. However, when we adjusted for leukocyte count, we found that smoking was weakly associated with MPO-DNA

**Table 2. Associations between MPO-DNA complex levels and clinical characteristics.**

| | Univariate mean difference (95% CI) | p-value | Multivariate Age and sex adjusted mean difference (95% CI) | p-value |
|---|---|---|---|---|
| Age (per 10 years increase) | -0.04 (-0.06;-0.02) | <0.01 | -0.04 (-0.06;-0.02)[a] | <0.01 |
| Sex (male versus female) | -0.03 (-0.06;0.01) | 0.14 | -0.02 (-0.06;0.01)[b] | 0.20 |
| Current smoking (current versus never) | -0.03 (-0.07;0.02) | 0.27 | -0.04 (-0.09;0.01) | 0.10 |
| BMI (per 10 kg/m$^2$ increase) | 0.03 (-0.02;0.07) | 0.25 | 0.02 (-0.02;0.07) | 0.28 |
| Systolic blood pressure (per 10 mmHg increase) | -0.01 (-0.02;0.00) | 0.06 | -0.01 (-0.01;0.00) | 0.21 |
| Hypertension | -0.04 (-0.08;-0.00) | 0.05 | -0.02 (-0.06;0.01) | 0.22 |
| Diabetes Mellitus | -0.01 (-0.03;0.01) | 0.23 | -0.01 (-0.02;0.01) | 0.24 |
| • History of CVD | -0.07 (-0.13;-0.02) | 0.01 | -0.07 (-0.13;-0.01) | 0.02 |
| • History of CHD | -0.10 (-0.17;-0.03) | <0.01 | -0.11 (-0.18;-0.03) | 0.01 |
| History of stroke | 0.01 (-0.08;0.10) | 0.82 | 0.03 (-0.06;0.12) | 0.57 |
| Antithrombotic medication | -0.02 (-0.06;0.02) | 0.34 | -0.01 (-0.05;0.04) | 0.76 |
| Lipid-reducing agents | -0.06 (-0.12;-0.01) | 0.02 | -0.07 (-0.12;-0.02) | 0.01 |
| ***Blood measurements*** | | | | |
| Total cholesterol (mmol/L) | -0.00 (-0.02;0.02) | 1.00 | 0.00 (-0.02;0.02) | 0.96 |
| HDL (mmol/L) | -0.07 (-0.12;-0.03) | <0.01 | -0.07 (-0.12;-0.02) | 0.01 |
| Glucose (mmol/L) | -0.00 (-0.02;0.01) | 0.48 | -0.00 (-0.02;0.01) | 0.48 |
| CRP (mg/L) | -0.00 (-0.00;0.00) | 0.47 | -0.00 (-0.00;0.00) | 0.70 |
| Leukocytes ($*10^{-9}$/L) | 0.01 (-0.00;0.02) | 0.14 | 0.01 (-0.00;0.01) | 0.31 |
| Fibrinogen (g/L) | -0.01 (-0.02;0.01) | 0.63 | 0.00 (-0.02;0.02) | 0.81 |
| VWF (IU/mL) | -0.00 (-0.03;0.03) | 0.84 | 0.01 (-0.02;0.04) | 0.49 |
| ADAMTS13 (%) | 0.00 (-0.00;0.00) | 0.41 | 0.00 (-0.00;0.00) | 0.78 |
| VWF/ADAMTS13 ratio | -1.06 (-3.16;0.65) | 0.20 | -0.36 (-2.37;1.65) | 0.73 |

[a]Age adjusted for sex.

[b]Sex adjusted for age. MPO-DNA complex levels were log-transformed. CVD: cardiovascular disease, CHD: coronary heart disease. HDL: high density lipoprotein, CRP: C-reactive protein, VWF: Von Willebrand Factor, ADAMTS13: a disintegrin and metalloproteinase with a thrombospondin type 1 motif, member 13.

complex levels (β -0.05 mAU/mL, 95% CI -0.10;-0.00). In participants in the age category 55–65 years, we found that smoking was weakly associated with MPO-DNA complex (β -0.08 mAU/mL, 95% CI -0.15;-0.01). In the age category 65–75 years, the use of lipid-lowering drugs was more important (β -0.09 mAU/mL, 95% CI -0.17;-0.01) and in participants >75 years, HDL levels (β -0.12 mAU/mL, 95% CI -0.21;-0.04) and a history of CHD (β -0.12 mAU/mL, 95% CI -0.23;0.00) were only weak determinants of MPO-DNA complex levels. When analyzing men and women separately, we found that a history of CHD was a weak determinant for MPO-DNA complex levels in men (β -0.14 mAU/mL, 95% CI -0.22;-0.05), but not in women (β -0.07 mAU/mL, 95% CI -0.21;0.08). On the other hand, MPO-DNA complex levels were weakly associated with HDL levels in women (β -0.09 mAU/mL/ mmol/L, 95% CI -0.14;-0.03), but not in men (β -0.03 mAU/mL/ mmol/L, 95% CI -0.11;0.05). Subgroup analysis in participants with and without any comorbidities, showed a weak inverse association between MPO-DNA complex levels and age (respectively, difference per 10 year increase -0.03 mAU/mL, 95% confidence interval (CI) -0.05;-0.00 and difference per 10 year increase -0.07 mAU/mL, 95% confidence interval (CI) -0.12;-0.02). When we added age, a history of CHD, the use of lipid-lowering drugs and levels of HDL to the same model, all variables remained significant, although the associations were weak.

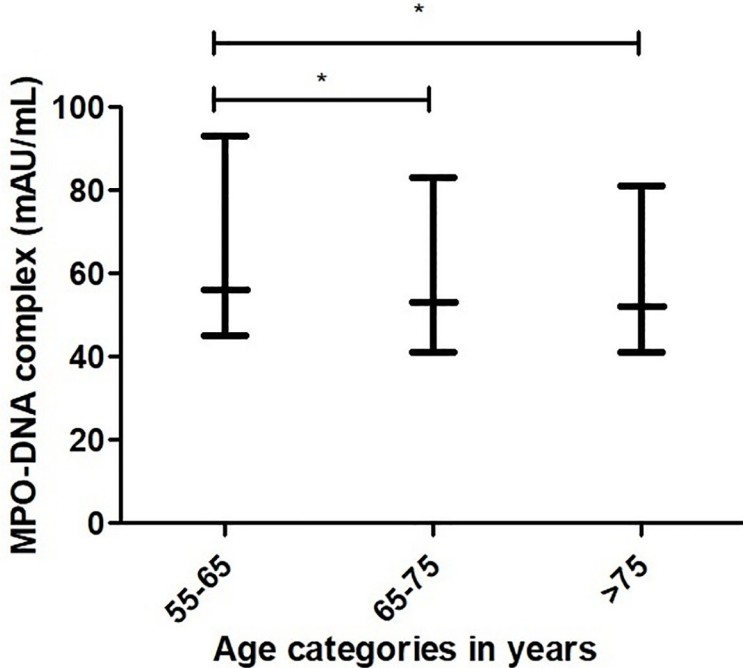

**Fig 1. Distribution of MPO-DNA complex levels among age categories.** *p<0.01. Data are presented as median and 25th-75th percentiles. Category 55–65 years, (n = 2286) MPO-DNA complex 56 mAU/mL (45–93), category 65–75 years (n = 2467) MPO-DNA complex 53 mAU/mL (41–83), category >75 years (n = 1674) MPO-DNA complex 52 AU/mL (41–81). Differences between age categories were analyzed using the Kruskall–Wallis test with post-hoc analysis.

## MPO-DNA complex plasma levels are not associated with inflammatory, immunology and coagulation markers

MPO-DNA complex levels were not associated with levels of C-reactive protein (CRP), Von Willebrand Factor (VWF), ADAMTS13, or fibrinogen (Table 2, S1 Table in S1 File). There was a weak correlation between NETs and leukocyte count ($R_s$ = 0.03, p = 0.02). However, in regression analyses, MPO-DNA complex levels were not associated with the concentration of leukocytes (β 0.01 mAU/mL /*10-9/L, 95% CI -0.00–0.02), indicating that the amount of NETs were not influenced by the number of leukocytes. In the exploratory analysis among the subset of 1208 participants in whom an extended panel of inflammatory markers was measured, MPO-DNA complex levels were weakly associated with TNFα (β 0.02 mAU/ml/ pg/mL, 95% CI 0.00–0.04) and IgM (β 0.11 mAU/mL/ g/L, 95% CI 0.05–0.17) (Table 3, S3 Table in S1 File).

## Discussion

In this population-based cohort study we found negative associations between MPO-DNA complex levels and age, HDL levels, the use of lipid-lowering drugs and a history of CHD. Although these associations were significant, the effects of these determinants on NETs levels were only mild. We found no associations with markers of inflammation, immunology or coagulation. To our knowledge, this is the first study that investigated the association between demographic and clinical characteristics and plasma markers of NET formation in the general population.

Although the effect was limited, we observed a small decrease in levels of MPO-DNA complex with advancing age, which was independent of the presence of comorbidities. In previous

**Table 3. Associations between MPO-DNA complex and inflammatory and immunology markers in a subset of 1208 individuals of RS-I-3.**

| | Univariate mean difference (95% CI) | p-value | Multivariate Age and sex adjusted mean difference (95% CI) | p-value |
|---|---|---|---|---|
| Complement factor C3 (g/L) | -0.01 (-0.25;0.24) | 0.97 | -0.00 (-0.25;0.25) | 0.98 |
| IgA (g/L) | -0.03 (-0.15;0.09) | 0.59 | -0.02 (-0.15;0.10) | 0.75 |
| IgE (g/L) | 0.00 (0.00;0.00) | 0.32 | 0.00 (0.00;0.00) | 0.26 |
| IgM (g/L) | 0.11 (0.05;0.17) | <0.01 | 0.11 (0.05;0.17) | <0.01 |
| IL-1beta (pg/mL) | 0.06 (-0.05;0.16) | 0.28 | 0.06 (-0.04;0.16) | 0.26 |
| IL-1ra (pg/mL) | -0.00 (-0.00;0.00) | 0.85 | -0.00 (-0.00;0.00) | 0.86 |
| IL-3 (pg/mL) | 0.03 (-0.48;0.54) | 0.91 | 0.04 (-0.47;0.55) | 0.89 |
| IL-4 (pg/mL) | -0.00 (-0.00;0.00) | 0.14 | -0.00 (-0.00;0.00) | 0.15 |
| IL-5 (pg/mL) | 0.00 (-0.00;0.00) | 0.61 | 0.00 (-0.00;0.00) | 0.60 |
| IL-7 (pg/mL) | 0.00 (-0.00;0.00) | 0.52 | 0.00 (-0.00;0.00) | 0.55 |
| IL-8 (pg/mL) | 0.00 (-0.00;0.01) | 0.53 | 0.00 (-0.00;0.01) | 0.49 |
| IL-10 (pg/mL) | -0.00 (-0.01;0.01) | 0.90 | -0.00 (-0.01;0.01) | 0.89 |
| IL-12p70 (pg/mL) | 0.00 (0.00;0.00) | 0.04 | 0.00 (0.00;0.00) | 0.46 |
| IL-13 (pg/mL) | -0.00 (-0.00;0.00) | 0.19 | -0.00 (-0.00;0.00) | 0.19 |
| IL-15 (pg/mL) | 0.07 (-0.14;0.28) | 0.53 | 0.07 (-0.15;-0.28) | 0.54 |
| IL-16 (pg/mL) | 0.00 (0.00;0.00) | 0.13 | 0.00 (0.00;0.00) | 0.10 |
| IL-17 (pg/mL) | 0.01 (-0.00;0.02) | 0.08 | 0.01 (-0.00;0.02) | 0.07 |
| IL-18 (pg/mL) | 0.00 (0.00;0.00) | 0.78 | 0.00 (0.00;0.00) | 0.83 |
| TNFα (pg/mL) | 0.02 (0.00;0.04) | 0.04 | 0.02 (0.00;0.04) | 0.04 |

MPO-DNA complex levels were log-transformed. IL: interleukin, Ig: immunoglobulin, TNF: tumor necrosis factor.

studies, it has been described that the occurrence of inflammatory diseases promotes NET formation. For instance, higher levels of NETs are found in type 2 diabetes mellitus [20], heart failure [21] and thrombosis [3]. Since the presence of comorbidities increase with age, it would be expected that also NETs levels increase with age. However, in this study we observed the opposite. An age-related decline of NET formation has also been described in several *in vivo* and *in vitro* studies. In mice, neutrophils from older mice exposed to methicillin-resistant Staphylococcus aureus, have lower levels of NETs than neutrophils isolated from young mice [9]. Accordingly, Xu et al. reported lower levels of NETs in aged mice and found that a defect in Atg5-related autophagy may contribute to this decrease [22]. Also in elderly patients with periodontitis, NET formation in neutrophils is lower than in younger controls [10]. It could be that in some diseases the NETs levels indeed increase compared to healthy controls, but that neutrophils lose the ability to form large amounts of NETs with increasing age, irrespective of the presence of comorbidities. It is known that the elderly have an overall increased susceptibility to infection and also have a suboptimal immune response after vaccination [23–25]. Immunosenescence is part of the aging process and also effects neutrophil function [26]. Since neutrophil function decreases with age, the formation of NETs could possibly also be decreased.

In this study, we found some weak associations between NETs and CVD risk factors, such as HDL, history of CHD and the use of lipid lowering drugs. The precise mechanisms behind these associations remain unknown. In case of HDL and lipid lowering drugs, a possible anti-inflammatory effect might play a role [27, 28]. In previous studies on patients with CVD, the role of NETs have already been well described. For instance, in patients with coronary atherosclerosis, high NETs levels were associated with major adverse cardiovascular events [4]. Also, in patients with an acute ST-elevation MI (STEMI), NETs are elevated in the acute phase [29]. The main difference between previous studies and our study, is that we measured NETs in a

community dwelling population which cannot directly be compared to case-control studies in patients with active or acute disease. In addition, in case-control studies, specific patient groups are selected with certain disease characteristics and disease severities. In this study, such a selection was not present and individuals with minor disease as well as severe disease were all included.

Since CRP is an acute phase protein and a marker of overall inflammation, we expected to find a relation between CRP and MPO-DNA complex levels, but surprisingly found no evidence of this in our population. Although NETs are also formed in several chronic diseases, high amounts of NETs are formed in an acute event [30]. In this first population study on NETs levels, only a small percentage of participants experienced a recent acute event. Most participants were in a steady inflammatory state, as can also be derived from the low CRP levels in this population. This could explain the low levels of MPO-DNA complex in this study. Also, leukocyte counts were not associated with MPO-DNA complex levels. This adds to a previous study performed in patients with an acute ST-elevation MI (STEMI), MPO-DNA complex levels only correlated with the total leukocyte count in the acute phase and not in the stable phase, 3 months after the event [29]. In an extended panel of inflammatory and immunology markers in a subset of participants, we found a positive association with TNFα which is known to be involved in innate immunity, and with an immunoglobulin linked to adaptive immunity (IgM). In a recent systematic review, TNFα was found to function as an inducer of NET formation in five out of seven studies [17]. However, there is no literature on IgM as a potential inducer of NET formation. The underlying mechanism driving the associations between IgM and MPO-DNA complexes, may be the result of neutrophil activation and subsequent initiation of adaptive immunity.

On the basis of prior studies [3, 31], we anticipated to find a link between NET formation and coagulation factors like VWF, ADAMTS13, and fibrinogen. VWF is released from endothelial cells as a result of NETs induced endothelial injury. VWF directly binds to the negatively charged DNA network of the NETs and thereby immobilizes NETs to the vessel wall, while at the same time platelets bind to the NETs and become activated, perpetuating the prothrombotic nature of NETs [3, 31]. One possible explanation for why we found no association with coagulation in our study, is the absence of an acute inflammatory state in this population. According to the 'immunothrombosis' hypothesis, NETs activate the coagulation system in response to blood-borne pathogens [32]. These conditions are present in only a small proportion of the general population, as evidenced by the low levels of CRP in our study.

Here, we measured MPO-DNA complex levels as marker for the presence of NETs in plasma. MPO-DNA complex is currently considered the most specific, objective and quantitative assay for monitoring NET formation [33]. These complexes are remnants of NETs and are formed during the process of NET formation, when MPO binds to nuclear DNA and synergizes with neutrophil elastase (NE) in decondensing chromatin [34]. Subsequently, intracellular DNA forms a complex with MPO and other antimicrobial proteins (i.e. NE) and is being released from the neutrophil to form NETs. Besides MPO-DNA complex, another widely used marker for NET formation is citrullinated histone 3 (CitH3). An important step for NET formation is decondensation of chromatin which is promoted by different proteins, including MPO and protein-arginine deiminase type 4 (PAD4) [35]. PAD4 is a nuclear enzyme that citrullinates arginine. CitH3 is a marker of this PAD4-dependent pathway of NET formation. In addition, phosphoinositide 3-kinase (PI3K) is also required for the formation of NETs, implicating the importance of the autophagy pathway [36]. This is supported by a study in promyelocytes that lack the autophagy-associated protein ATG7, where a decrease in NET release was observed [37]. Where CitH3 is only a marker for the PAD4-dependent pathway of the formation of NETs, MPO-DNA complex also represents the autophagy pathway. In a subset of

participants, we measured both MPO-DNA complex and CitH3 to investigate the correlation between the two markers in plasma. However, we found that MPO-DNA complex and CitH3 were not correlated in a small subset of the general population where there is no acute inflammation (S3 Fig in S1 File).

The main strength of this study, is that this is the first study that measured MPO-DNA complex levels in a very large community dwelling population and investigated the association with several known CVD risk factors. Although we were unable to demonstrate a clinical relevant association between NETs and any of the CVD risk factors, the findings of this study are still of importance to identify the role of NETs on population level. A possible limitation of this study is that most participants had very low levels of MPO-DNA complex (<100 mAU/mL). This limited variability might have hampered the identification of determinants of NET formation in this population. Also, the differences in MPO-DNA complex levels between participants with and without prevalent CVD were small. However, in previous studies on CRP levels and the risk of CVD, small elevations in CRP levels within the normal reference range have been shown to be associated with CHD [38–40]. Thus, this indicates that low grade inflammation is a risk factor for CHD. We therefore hypothesized that low levels of NETs, within the normal range, may be of biological relevance. Another limitation of this study is the possibility of confounding factors. For this study, we were interested in identifying determinants of NETs levels. When adjusting for possible confounders, there is a risk of overadjusting, resulting in the inability to identify possible determinants. This in in contrast to studies focusing on disease risk, where it is customary to adjust for possible confounders. Although the associations we found in this study were significant, it is doubtful that these associations are of clinical relevance. This may suggest that NETs are more important in the acute phase of disease than in a steady state situation. Future studies focusing on determining reference values of MPO-DNA complex should be conducted to investigate clinical relevance of given values.

## Conclusions

Older age, a history of coronary heart disease, the use of lipid-lowering drugs and higher HDL-cholesterol are weakly correlated with lower plasma levels of NETs. The findings of this study demonstrate that the effect of CVD risk factors on NETs levels in a general population is limited and may not be of clinical relevance. This emphasizes that NETs may play a more important role in an acute phase of disease than in a steady state situation.

## Supporting information

**S1 File. Supplemental tables and figures.** S1 Table. Correlations between MPO-DNA complex and clinical characteristics. S2 Table. MPO-DNA complex levels in CVD risk factors. S3 Table. Correlations between MPO-DNA complex and inflammatory and immunology markers in a subset of 1208 individuals of RS-I-3. S1 Fig. Circadian rhythm of MPO-DNA complex levels during daytime. S2 Fig. Typical example of a reference curve used for MPO-DNA complex ELISA. S3 Fig. Correlation between MPO-DNA complex and citrullinated histone H3. (DOCX)

## Acknowledgments

The authors would like to acknowledge J.W.R van Soerland and F. Dik for their excellent help with the measurement of MPO-DNA complex. The contribution of inhabitants, general

practitioners and pharmacists of the Ommoord district to the Rotterdam Study is gratefully acknowledged.

## Author Contributions

**Conceptualization:** M. Arfan Ikram.

**Formal analysis:** Samantha J. Donkel, Frank J. Wolters.

**Methodology:** M. Arfan Ikram, Moniek P. M. de Maat.

**Supervision:** M. Arfan Ikram, Moniek P. M. de Maat.

**Writing – original draft:** Samantha J. Donkel.

**Writing – review & editing:** Frank J. Wolters, M. Arfan Ikram, Moniek P. M. de Maat.

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
