## [Decision Letter · Decision Letter 0]

26 Mar 2021

PONE-D-20-40069

Circulating neutrophil extracellular traps (NETs) levels and cardiovascular risk factors in the general population

PLOS ONE

Dear Dr. Donkel,

Thank you for submitting your manuscript to PLOS ONE. After careful consideration, we feel that it has merit but does not fully meet PLOS ONE’s publication criteria as it currently stands. Therefore, we invite you to submit a revised version of the manuscript that addresses the points raised during the review process.

Two experts in the field have reviewed the study that, while considering the interesting results provided, find that the conclusions are only partly supported by the data. In part, this is attributed to the fact that studying the general population, no strong association is to be expected. As indicated by one of the reviewers, this fact should be used as both a limitation and a strength of the study. In general, the discussion seems to be unnecessarily long. The authors should indicate in the title that the study measures MPO-DNA.  As this is a surrogate marker of NETosis and not a direct measurement of NETs, the title could be misleading.

We look forward to receiving your revised manuscript.

Kind regards,

Pablo Garcia de Frutos

Academic Editor

PLOS ONE

Journal Requirements:

'The Rotterdam Study is supported by the Erasmus MC University Medical Center and Erasmus University Rotterdam; The Netherlands Organisation for Scientific Research (NWO); The Netherlands Organisation for Health Research and Development (ZonMw); the Research Institute for Diseases in the Elderly (RIDE); The Netherlands Genomics Initiative (NGI); the Ministry of Education, Culture and Science; the Ministry of Health, Welfare and Sports; the European Commission (DG XII); and the Municipality of Rotterdam. We acknowledge the support of the Netherlands Cardiovascular Research Initiative which is supported by the Dutch Heart Foundation (CVON2015-01: CONTRAST), the support of the Brain Foundation Netherlands (HA2015.01.06), and the support of **Health~Holland**, Top Sector Life Sciences & Health (LSHM17016),** Medtronic and Cerenovus** . The collaboration project is additionally financed by the Ministry of Economic Affairs by means of the PPP Allowance made available by the Top Sector Life Sciences & Health to stimulate public-private partnerships. The measurement of MPO-DNA complex levels in participants of the Rotterdam study was supported by a research grant (Prof. Heimburger Award 2018, CSL Behring).'             

6. Thank you for stating the following in the Financial Disclosure section:

'The Rotterdam Study is supported by the Erasmus MC University Medical Center and Erasmus University Rotterdam; The Netherlands Organisation for Scientific Research (NWO); The Netherlands Organisation for Health Research and Development (ZonMw); the Research Institute for Diseases in the Elderly (RIDE); The Netherlands Genomics Initiative (NGI); the Ministry of Education, Culture and Science; the Ministry of Health, Welfare and Sports; the European Commission (DG XII); and the Municipality of Rotterdam. We acknowledge the support of the Netherlands Cardiovascular Research Initiative which is supported by the Dutch Heart Foundation (CVON2015-01: CONTRAST), the support of the Brain Foundation Netherlands (HA2015.01.06), and the support of Health~Holland, Top Sector Life Sciences & Health (LSHM17016),** Medtronic and Cerenovus**. The collaboration project is additionally financed by the Ministry of Economic Affairs by means of the PPP Allowance made available by the Top Sector Life Sciences & Health to stimulate public-private partnerships. The measurement of MPO-DNA complex levels in participants of the Rotterdam study was supported by a research grant (Prof. Heimburger Award 2018, CSL Behring).'

We note that you received funding from a commercial source: Health-Holland, Medtronic and Cerenovus

b.Please include your amended Competing Interests Statement within your cover letter. We will change the online submission form on your behalf.

Reviewers' comments:

Reviewer's Responses to Questions

**Comments to the Author**

1. Is the manuscript technically sound, and do the data support the conclusions?

Reviewer #1: Partly

Reviewer #2: Partly

2. Has the statistical analysis been performed appropriately and rigorously? 

Reviewer #1: No

Reviewer #2: I Don't Know

3. Have the authors made all data underlying the findings in their manuscript fully available?

Reviewer #1: Yes

Reviewer #2: Yes

4. Is the manuscript presented in an intelligible fashion and written in standard English?

Reviewer #1: Yes

Reviewer #2: Yes

5. Review Comments to the Author

Reviewer #1: In this study, the authors aimed to assess whereas neutrophil extracellular trap (NET) formation is associated with demographic and/or cardiovascular risk factors in general population. NETs were quantified by measuring MPO-DNA complex using an ELISA. The results showed that MPO-DNA complex levels were lower with advancing age, a history of coronary heart disease, the use of lipid-lowering drugs and higher HDL-cholesterol. Although the pathophysiologic hypothesis is intriguing, this reviewer has some concerns:

- The differences in MPO-DNA complex levels between subjects with and without cardiovascular risk factors were very small and the biological and clinical relevance seems forced.

- In the Methods section, the definition of cardiovascular risk factors seems partial and incomplete, mainly for hypertension and hypercholesterolemia. The pharmacological therapy was only partially considered and the definition of “coronary heart disease” seems insufficient.

- The model used for multivariable regression analysis only contemplate age and sex adjustment while other variables with a p value <0.1 at univariable analysis were not considered. Furthermore, there were no significant differences in MPO-DNA complex levels between male and female at univariate analysis.

- The discussion paragraph is too long. The argument is complex and sometimes farfetched and vulnerable.

Reviewer #2: In this manuscript, Donkel et al. evaluated the levels of NETs in a large cohort from the prospective population-based Rotterdam Study. The authors investigated the association between NET levels, measured by MPO-DNA ELISA, and different clinical, biochemical, cellular and inflammatory parameters. The study is certainly of interest given that this kind of study is missing and may give important answers on NET pathophysiology.

The main criticism that I have is on how the results are presented. The authors should be cautious with the conclusions. All the correlations are at most weak (max Rs is 0.11!) and the majority are very weak (<0.1). Thus, as the authors pointed it out in the limitation section, the biological, pathophysiological, and clinical relevance of these associations are very difficult to establish. The presence of confounding variables adds additional difficulty to interpret data.

The authors should not be scared to present the manuscript focusing on that NETs are not or very weakly associated with the different parameters since this study is very informative regarding a healthy population. In this sense, the abstract gives a message that is not conveyed once the manuscript is read. In particular, the conclusions of the abstract and of the manuscript should be rephrased and soften. The discussion should be reduced since several very weak associations do not need so much discussion.

This work has to show its strengths, that are the casuistic and the large amount of parameters to perform the statistics. The authors may give conclusions concerning the lack of associations more than the presence of associations between NETs and different parameters.

Beside this main concern, other minor points should be attended:

• Pleas e check for typo errors (e.g. line 117: phorbol 12-myristaat 13-acetaat)

• The concentration of PMA used to activate NETosis has to be indicated in M&M

• Line 191, please add the reference to Figure 1

• Line 313. Authors should moderate the sentence given that other studies have shown a correlation between MPO/DNA and cfDNA (e.g. 31119471 (septic patients), 32329756 (Covid-19))

• Line 326. This result is important and should be shown.

• Significant correlations have to be shown in a graph.

• How do the authors explain the lower levels of NETs in older subjects? Can this be associated with drug intake that can be higher in older vs younger individuals?

6. PLOS authors have the option to publish the peer review history of their article (what does this mean?). If published, this will include your full peer review and any attached files.

Reviewer #1: No

Reviewer #2: No

---

## [Author Response · Author response to Decision Letter 0]

24 May 2021

Response to the reviewers was uploaded in a separate file called 'response to the reviewers'.

---

## [Decision Letter · Decision Letter 1]

11 Jun 2021

Circulating Myeloperoxidase (MPO)-DNA complexes as marker for neutrophil extracellular traps (NETs) levels and the association with cardiovascular risk factors in the general population

PONE-D-20-40069R1

Dear Dr. Donkel,

We’re pleased to inform you that your manuscript has been judged scientifically suitable for publication and will be formally accepted for publication once it meets all outstanding technical requirements.

Kind regards,

Pablo Garcia de Frutos

Academic Editor

PLOS ONE

Additional Editor Comments (optional):

Reviewers' comments:

Reviewer's Responses to Questions

**Comments to the Author**

1. If the authors have adequately addressed your comments raised in a previous round of review and you feel that this manuscript is now acceptable for publication, you may indicate that here to bypass the “Comments to the Author” section, enter your conflict of interest statement in the “Confidential to Editor” section, and submit your "Accept" recommendation.

Reviewer #1: All comments have been addressed

Reviewer #2: All comments have been addressed

2. Is the manuscript technically sound, and do the data support the conclusions?

Reviewer #1: Yes

Reviewer #2: Yes

3. Has the statistical analysis been performed appropriately and rigorously? 

Reviewer #1: Yes

Reviewer #2: Yes

4. Have the authors made all data underlying the findings in their manuscript fully available?

Reviewer #1: Yes

Reviewer #2: No

5. Is the manuscript presented in an intelligible fashion and written in standard English?

Reviewer #1: Yes

Reviewer #2: Yes

6. Review Comments to the Author

Reviewer #1: The manuscript has been somewhat improved.

Despite some limitations and its weak results, the study remains of scientific interest.

Reviewer #2: (No Response)

7. PLOS authors have the option to publish the peer review history of their article (what does this mean?). If published, this will include your full peer review and any attached files.

Reviewer #1: No

Reviewer #2: No

---

## [Editor Report · Acceptance letter]

2 Aug 2021

PONE-D-20-40069R1 

Circulating Myeloperoxidase (MPO)-DNA complexes as marker for neutrophil extracellular traps (NETs) levels and the association with cardiovascular risk factors in the general population 

Dear Dr. Donkel:

I'm pleased to inform you that your manuscript has been deemed suitable for publication in PLOS ONE. Congratulations! Your manuscript is now with our production department. 

Kind regards, 

on behalf of

Dr. Pablo Garcia de Frutos 

Academic Editor

PLOS ONE